# Early Valve Replacement for Severe Aortic Valve Disease: Effect on Mortality and Clinical Ramifications

**DOI:** 10.3390/jcm9092694

**Published:** 2020-08-20

**Authors:** Jason P. Koerber, Jayme S. Bennetts, Peter J. Psaltis

**Affiliations:** 1Department of Anaesthesia, Flinders Medical Centre, Adelaide, SA 5042, Australia; Jason.Koerber@sa.gov.au; 2Department of Cardiac and Thoracic Surgery, Flinders Medical Centre, Adelaide, SA 5042, Australia; Jayme.Bennetts@sa.gov.au; 3College of Medicine and Public Health, Flinders University, Adelaide, SA 5042, Australia; 4Adelaide Medical School, Faculty of Health and Medical Sciences, University of Adelaide, Adelaide, SA 5000, Australia; 5Vascular Research Centre, Lifelong Health Theme, South Australian Health and Medical Research Institute, Adelaide, SA 5000, Australia; 6Department of Cardiology, Central Adelaide Local Health Network, Adelaide, SA 5000, Australia

**Keywords:** valvular heart disease, aortic valve replacement, transcatheter aortic valve implantation, survival, aortic regurgitation, aortic stenosis, mixed aortic valve disease

## Abstract

Timing of aortic valve intervention for chronic aortic regurgitation (AR) and/or aortic stenosis (AS) potentially affects long-term survival. The 2014 American Heart Association/American College of Cardiology (AHA/ACC) guidelines provide recommendations for the timing of intervention. Subsequent to the guidelines’ release, several studies have been published that suggest a survival benefit from earlier timing of surgery for severe AR and/or AS. The aim of this review was to determine whether patients who have chronic aortic regurgitation (AR) and/or aortic stenosis (AS) have a survival benefit from earlier timing of aortic valve surgery. Medical databases were systematically searched from January 2015 to April 2020 for randomized controlled trials (RCTs) and observational studies that examined the timing of aortic valve replacement surgery for chronic AR and/or AS. For chronic AR, four observational studies and no RCTs were identified. For chronic AS, five observational studies, one RCT and one meta-analysis were identified. One observational study examining mixed aortic valve disease (MAVD) was identified. All of these studies, for AR, AS, and MAVD, found long-term survival benefit from timing of aortic valve surgery earlier than the current guidelines. Larger prospective RCTs are required to evaluate the benefit of earlier surgical intervention.

## 1. Introduction

The timing of surgical intervention is an immensely important decision for patients who have significant aortic valve disease. Aortic valve surgery (AVS), by replacement (AVR) or repair (AVr) is the only form of treatment that can substantially alter the dismal natural history of severe aortic regurgitation (AR) or severe aortic stenosis (AS) [1]. Open heart surgery used to be the only option with its risk of peri-operative mortality and morbidity, and either ongoing thromboembolic/anticoagulation risk for mechanical valves or finite durability before bioprosthetic valves or repairs fail [2]. Now, for an increasing proportion of patients, transcatheter aortic valve replacement (TAVI) offers a less invasive method, but with comparable periprocedural risks, ongoing thromboembolic risks, and an unclear long-term durability [3]. Patients are concerned about the morbidity and mortality risks of valve intervention but also want to avoid the poor long-term consequences of uncorrected aortic valve disease. They wish to maintain or improve their level of activity and quality of life. When clinicians provide timing advice to assist their patients, they are required to balance these factors while also considering the patient’s age, comorbidities, lifestyle and wishes.

Severe AS is associated with increased mortality, as shown by a recent large Australian echocardiographic database study [4] involving 241,303 individuals aged ≥18 years who presented for an echocardiogram for any reason. This study found that the presence of severe AS increased 5 year mortality by 3.0-fold compared to when no AS was present (*p* < 0.001). This marked difference persisted after adjusting for age, sex, left ventricular systolic or diastolic dysfunction, and AR. This shows that despite contemporary medical practice, AS is associated with a loss in life expectancy. Even following AVR, there is an associated loss of life expectancy, as shown by a Swedish national registry study [5] that included 23,528 patients who underwent primary surgical AVR between 1995 and 2013. It found that after a mean follow-up period of 6.8 years, the relative survival of patients following AVR was 63% compared to matched general population. Notably, the long-term survival following surgical correction for AR is worse than AS [6,7]. In light of these sobering data, there is still much potential room for long-term mortality improvements for both AS and AR.

Until recently, there had been no published randomized controlled trial (RCT) exploring the timing of AVR surgery [8]. The only evidence base available was drawn from observational and mostly retrospective data [9,10]. Non-randomized data are particularly prone to bias, so it is a challenge to confidently make clinical recommendations for the timing of aortic valve surgery.

The most recent American guidelines regarding the timing of aortic valve surgery are the 2014 American Heart Association/American College of Cardiology (AHA/ACC) guidelines for the management of patients with valvular heart disease [11]. There was a focused 2017 AHA/ACC update that altered recommendations for choice of intervention (TAVI versus surgical AVR) but did not alter intervention timing recommendations [12]. The 2014 AHA/ACC guidelines in combination with the equivalent European guidelines [13] provide a framework for clinicians to make evidence-based decisions regarding timing of AVR. Since their release, several studies have been published that examine timing of surgery for either AR or AS [8,14,15,16,17,18,19,20,21,22,23]. These studies suggest a possible survival benefit from earlier timing of AVR. Therefore, this review investigates the strength of these additional data, how they relate to the data that the 2014 guidelines were built upon, and whether the guidelines’ recommendations warrant revisiting and updating.

## 2. Literature Search and Information Sources

The evidence summarized and presented in this review arises from digital literature searches conducted through MEDLINE, Embase and Google Scholar. A flowchart of the literature selection is presented in Figure 1. Titles and abstracts found through the searches were screened for relevance. Further searches were constructed from relevant articles’ reference lists and citing articles. All articles referenced by the AHA/ACC 2014 guidelines relating to timing of AVR were also searched.

A statistical meta-analysis was not performed because the data were very heterogenous. However, all results pointed in the same direction and most individual articles achieved statistical significance.

## 3. AHA/ACC 2014 Guidelines

Key recommendations from the 2014 guidelines regarding timing of AR and AS intervention are summarized in Table 1 and Table 2. For AR, the main triggers for intervention are the development of symptoms, left ventricular systolic dysfunction (left ventricular ejection fraction [LVEF] <50%), left ventricular end-systolic dimension (LVESD) dilatation (LVESD >50 mm, or LVESD indexed to body surface area (LVESDi) >25 mm/m^2^) and concomitant cardiac surgery. For AS, three of the main triggers are similar to AR: development of symptoms, left ventricular systolic dysfunction (LVEF <50%) and concomitant cardiac surgery. Other triggers for severe AS are very severe aortic velocity (≥5.0 m/s), abnormal exercise stress test results, low-flow severe AS, and with the lowest level evidence (IIb), may be considered for asymptomatic patients with severe AS, rapid disease progression and low surgical risk.

The 2014 guidelines with the focused 2017 update, draw no distinction between timing for TAVI or open-heart surgery. This remains relevant considering subsequently published trials have found similar survival curves between TAVI and surgical AVR for patients who have high, medium or low surgical risk [24,25,26,27]. The lower morbidity from TAVI procedures does open the possibility for intervention for patients who previously would have been considered as poor open-heart surgical candidates.

The latest European guidelines for timing of aortic valve surgery are the 2017 European Society of Cardiology/European Association for Cardio-Thoracic Surgery (ESC/EACTS) guidelines for the management of valvular heart disease [13]. The ESC/EACTS recommendations were essentially the same as the AHA/ACC guidelines so for brevity, this review has chosen to concentrate solely on the 2014 AHA/ACC guidelines.

## 4. Aortic Regurgitation: Studies Since 2015

Four observational studies and no RCTs were identified that examined the timing of surgery for chronic AR and were published since 2015, summarized in Table 3 [14,15,16,17]. These four studies had in common that they were retrospective, observational studies and used at least 10 year survival as their primary outcome. Two of the studies recruited only patients who underwent AVR [15,17], while the other two studies involved patients who received either AVR or conservative management during their study periods [14,16]. The two studies with cohorts split between AVR and conservative management, both found significantly better survival in the patients who underwent AVR. One of these two studies [14] compared subgroups where patients either did not or did meet the guideline-recommended LV dimension cut-offs for AVR at the time of surgery. The patients who had AVR before their LV dimensions had reached the guidelines’ recommended cut-offs had significantly better survival than those whose LV dimensions did reach the guidelines’ recommendations.

All four studies found that the presence of symptoms was associated with significantly poorer survival compared to the absence of symptoms. This is notable because the current guidelines do not recommend surgery until symptoms have developed.

Preoperative LVEF status was shown to be significantly associated with survival when LVEF was ≥60% [15] and ≥55% [17]. This contrasts with the guidelines’ recommendation of waiting until LVEF falls below 50%.

Yang et al. 2019 [16] found LVESDi to be the best left ventricular chamber predictor of survival with survival decreasing once LVESDi was greater than 20 mm/m^2^ and further decreasing once LVESDi was greater than 25 mm/m^2^. Similarly, de Meester et al. [17] found survival to be worse when LVESDi ≥25 mm/m^2^ compared to <25 mm/m^2^. These studies suggest a survival benefit from surgery prior to the current guidelines’ recommendation of waiting until LVESDi is greater than 25 mm/m^2^.

To summarize, the authors of these more recent studies advocated considering surgery for chronic AR earlier than the current guidelines’ recommendations. They suggested that long-term survival could be improved by performing surgery while patients are still asymptomatic, have an LVESDi of 20 to 25 mm/m^2^, and an LVEF >55–60%.

## 5. Aortic Stenosis: Studies Since 2015

Five observational studies, one RCT and one meta-analysis were identified that examined the timing of AVR for chronic AS published since the 2014 AHA/ACC guidelines (Table 4) [18,19,20,21,22,23]. All of these studies, except Lancellotti et al. (2018) [22], compared the survival of either an early AVR strategy or a guideline based ‘watchful waiting’ strategy. All of the studies exclusively recruited asymptomatic patients who had severe AS. Early AVR was consistently associated with significantly improved long-term survival compared to those managed with conservative ‘watchful waiting’ [8,18,19,20,21,23]. Of particular note, the RECOVERY trial [8] was the first published RCT investigating the timing of surgery for chronic AS. This small trial randomized 145 asymptomatic patients with severe AS (aortic valve area ≤0.75 cm^2^ with either an aortic jet velocity of ≥4.5 m/s or mean transaortic gradient of ≥50 mm Hg) to early surgical AVR or guideline-based watchful waiting with a median follow-up of 6.2 years. The early surgery group had significantly fewer deaths from cardiovascular or perioperative causes than the guideline-based group (1% vs. 15%, *p* = 0.003). All-cause death was also lower in the early surgery group (7% vs. 21%) [8].

Not only did these studies examine AVR before patients became symptomatic, the majority of the patients had a LVEF above the guideline recommended cut-off value of 50%. Four of the studies [8,20,22,23] involved recruiting only patients who had a LVEF ≥50%, while the other two studies [18,21] involved patients who had a mean LVEF of greater than 60%.

Rather than looking at the effect of an early AVR strategy, Lancellotti et al. (2018) [22] examined very severe peak aortic velocity (≥5.0 m/s) and also a LVEF cut-off value of ≥60%. They found that patients who had a preoperative peak aortic velocity ≥5.0 m/s had significantly worse survival compared to those with a peak aortic velocity <5.0 m/s. Patients who had surgery when LVEF was ≥60% survived significantly longer than when LVEF was <60%.

Overall, surgical AVR in asymptomatic severe AS patients with preserved LV function resulted in significant long-term survival benefits, suggesting that earlier AVR may be warranted before symptom development or ventricular dysfunction.

## 6. Perioperative Mortality

Perioperative mortality (also called operative mortality) is defined as deaths occurring during index surgery until 30 days postoperatively [28]. Over the last 30 years, there has been approximately a 10-fold decrease in perioperative aortic valve surgical mortality reported by observational studies to below 1% (Table 5) [7,14,16,22,23,28,29]. Recent national registries have reported comparable low mortality rates for isolated AVR; 1.9% in the U.S. [30], 1.01% in the U.K. [31], and 1.2% in Australia. [32]. This shifts the focus from perioperative mortality to long-term mortality.

## 7. Relating Data Since 2015 to Previous Data

The studies since 2015 that examined the timing of AVR for chronic AR or AS were compared with studies referenced by the AHA/ACC 2014 guidelines.

### 7.1. Chronic AR

Studies that were referenced by the AHA/ACC 2014 guidelines regarding the timing of AVR for chronic AR are summarized in the Appendix A [33,34,35,36,37,38,39,40,41,42,43,44,45,46]. All of these studies involved patients who had severe (grade ≥3) AR with marked left ventricular dilatation.

#### 7.1.1. Symptoms

The earlier studies showed that the presence of definite preoperative symptoms (NYHA functional class III/IV) was associated with reduced survival [39,44]. Even the presence of mild symptoms (class II versus class I) may reduce survival [39]. These studies suggest a survival benefit for timing surgery before symptoms arise. This is similar to the findings of the subsequent studies since 2015.

#### 7.1.2. LVEF

The guidelines referenced studies used LVEF cut-off values of 30% [33], 45% [35,42], 50% [36,41], 55% [39] and 58% [43]. For each LVEF cut-off value, survival benefit was found for higher preoperative LVEF. The earlier studies’ survival benefits for LVEF cut-off values as high as 55% and 58% were similar to the post-2015 data which showed benefits for LVEF cut-off values of 55% and 60% [16,17,39,43].

#### 7.1.3. LVESD/LVESDi

Three studies referenced by the AHA/ACC guidelines that considered LVESD or LVESDi, showed survival advantages when preoperative LVESD was <50 mm or LVESDi <25 mm/m^2^ [34,39,45]. Two other studies used a higher LVESD cut-off value of 55 mm, showing benefit when LVESD was <55 mm [37,43]. One small study had 100% survival at a mean of 44 months regardless of whether LVESD was greater than or less than 55 mm [40]. Overall, these references suggest that there may be a survival advantage when surgery occurs before LVESD rises above 50 mm or LVESDi above 25 mm/m^2^. Again, this is reproduced with the subsequent studies published since 2015.

#### 7.1.4. LVEDD/LVEDDi

Regarding left ventricular end-diastolic dimension (LVEDD) or indexed LVEDD (LVEDDi), the referenced studies did not reach statistical significance comparing LVEDD ≥70 mm against <70 mm [34,45]. The authors acknowledged the paucity of evidence and made an expert recommendation of AVR timing when LVEDD >65 mm. The subsequent studies since 2015 showed a similar lack of mortality discrimination based on LVEDD or LVEDDi.

#### 7.1.5. Chronic AR Summary Comparing New to Existing Data

For chronic AR, there is a consistent theme that the newer studies produced similar findings to the older studies. What has changed are the authors’ recommendations based on these data. The authors of the older studies and the 2014 AHA/ACC guidelines all recommended delaying intervention until patients had entered worse long-term survival categories. The authors of the more recent studies have recommended earlier surgery, and potentially before onset of symptoms, before patients reach the worse long-term survival categories. This change may be a result of greater confidence from the authors regarding their data’s veracity combined with the context of the large decrease in surgical perioperative mortality.

### 7.2. Chronic AS

The majority of the 2014 guidelines’ referenced studies that investigated AVR timing for AS, examined event-free survival [47,48,49,50,51,52,53,54,55,56], defined as alive without aortic valve surgery. These studies showed that there was rapid progression of asymptomatic patients who had severe AS, to becoming symptomatic and receiving aortic valve surgery. However, there were few deaths (<1%) while patients were asymptomatic [47,50,52,55], forming the basis for the recommendation that surgery can be safely delayed until symptoms occur. This assumes that preoperative symptom status does not affect long-term survival following AVR. This was supported by a small guideline referenced study (*n* = 128) that found no difference in 5 year survival following AVR for AS compared to matched general population [52]. However, the guidelines also referenced a contrasting larger study [7]. This Swedish (*n* = 2359) single-center, observational study found that patients who were asymptomatic or minimally symptomatic preoperatively, had better long-term survival following AVR than patients who had preoperative symptoms, with a mortality difference of approximately 20% at 10 and 15 years [7]. Despite the relative survival difference, the authors of this study were not willing to recommend earlier surgery based on their data.

The studies published subsequent to 2015 used long-term survival rather than event-free survival as their primary endpoint. They found a postoperative survival advantage when surgery occurred while patients were asymptomatic. Furthermore, the authors backed their data by recommending that surgery should be considered before patients become symptomatic.

#### LVEF

The guidelines recommend surgery when LVEF falls below 50%. The guidelines’ references did not directly examine this cut-off value’s effect on long-term mortality. Only one examined the effect of preoperative LVEF on postoperative mortality [22]. This study found improved survival when LVEF was greater than 60%. Further data would be helpful to clarify the relationship between preoperative LVEF and long-term survival for AS.

## 8. Features in Common between AR and AS and Possible Mechanisms

The newer studies suggest that once chronic severe AR or AS cause symptoms or left ventricular dysfunction, long-term survival decreases despite AVR. This implies that preoperative myocardial performance affects survival. One likely mechanism is irreversible myocardial damage. A histological study by Hein et al. 2003 explored the myocardial changes that occur in severe AS [57]. It showed that once myocyte degeneration and fibrosis have begun, a self-perpetuating process of myocyte degeneration, cell death and replacement fibrosis will be maintained even after AVR. This chronic cycle will lead to further impairment of left ventricular function and poor prognosis.

A more recent study by Chin et al. 2017 determined that cardiac magnetic resonance (CMR) can detect ventricular decompensation in AS through the identification of myocardial extracellular expansion and replacement fibrosis [58]. Based upon CMR results, they categorized patients into three levels of myocardial fibrosis: (1) normal myocardium; (2) extracellular expansion; (3) replacement fibrosis. The CMR changes correlated well with concurrent histological samples. All-cause mortality rate (per 1000 patient years) increased as the level of CMR fibrosis increased; 8 for normal myocardium, 36 for extracellular expansion and 71 for replacement fibrosis, *p* = 0.009. Their data showed that 46% of patients who had replacement fibrosis had no symptoms (NYHA class I), while 32% had mild symptoms (NYHA class II) [58]. LVEF was not able to predict patients’ level of fibrosis; LVEF was 67 (63–69), 66 (63–70) and 67 (63–72) for the three levels of fibrosis. This study provided evidence of the association of fibrosis with mortality in AS and also demonstrated that severe fibrosis frequently occurs before the onset of symptoms or fall in LVEF.

The idea that earlier surgery can increase longer-term survival by reducing myocardial damage, is not new. In 1990, Lund examined 5 year and 10 year survival following AVR for AS [59]. Using a Cox proportional hazards model, he showed that the improved survival over the period of 1965 to 1986, was related to improved preoperative patient status. Later patients had surgery with lower NYHA class scores. Lund hypothesized that the benefit to earlier surgery was probably predominantly related to less preoperative myocardial damage that caused later predictable death from congestive heart failure.

The development of symptoms or ventricular dysfunction may be a marker that permanent myocardial damage has already occurred. Earlier surgery has the potential to avoid permanent myocardial damage, leading to improved long-term survival.

## 9. Mixed Aortic Valve Disease

Mixed aortic valve disease (MAVD) is defined as simultaneous occurrence of AS and AR. There is clear evidence that moderate MAVD (moderate AS and moderate AR) progresses at a rate faster than isolated moderate AS or AR, with a progression rate similar to isolated severe AS [60,61,62,63,64,65,66]. A potential consequence of the rapid progression rate of moderate MAVD is that the window to identify and deal with significant lesions before significant irreversible myocardial damage has occurred, may be shorter. MAVD is characterized by a combination of pressure and volume load that imposes a greater stress on the left ventricle than that induced by isolated AS or AR [65]. When regurgitation is predominant, the pressure load from stenosis tends to restrict LV dilatation from the regurgitation’s volume load. When stenosis is predominant, the volume load from regurgitation will tend to exacerbate the pressure load from the stenosis.

Traditionally, the individual’s dominant lesion (stenosis or regurgitation) is compared against the criteria for an isolated stenosis/regurgitation [67]. Our search identified only one study that examined the effect of AVR timing on long-term survival for MAVD, a recently published observational cohort study that evaluated 862 patients (median 68 years old; 57% male) with preserved LVEF and at least moderate AR and moderate AS [68]. The cohort was divided into those who received medical management throughout the observation period (*n* = 357) and those who received an AVR (*n* = 505). Both groups displayed poor long-term survival, although AVR significantly reduced this risk; survival after a median follow-up period of 5.6 years was 31.9% in the medical management group and 64.8% in the AVR group (*p* < 0.001). However, comparison between the two groups was confounded by the AVR group at baseline being significantly younger, higher proportion male, higher proportion bicuspid and possessing less comorbidities. After propensity-matched subgrouping the large survival benefit for AVR persisted (*p* < 0.001). AVR improved survival regardless of symptoms or potentially modifying factors, such as smaller aortic valve area. The authors also found that peak (not mean) aortic valve gradients >45 mm Hg were associated with poor prognosis in the patients who were in the medical management group. This led to the authors concluding that a peak aortic valve gradient of 45 mm Hg could be used as a cut-off value for AVR. Considering that a peak aortic valve gradient of 45 mm Hg corresponds to a mean gradient well below 40 mm Hg, this is earlier than guideline recommendations for AS. While this study is consistent with the narrative from AR and AS that early AVR may produce a survival benefit, further studies are required to clarify whether and how the timing of AVR should be adjusted in MAVD. However, it is clear that vigilance is required.

## 10. Ramifications of Earlier Surgery

An early AVR strategy has to be balanced against the increased long-term complications from intervention. The rapid development of TAVI has increased the options available for patients contemplating earlier AVR but wishing to avoid lifelong anticoagulation from a mechanical valve.

### 10.1. Limited Durability of Bioprosthetic Valves or Risks from Anticoagulation

Earlier surgery may improve long-term mortality but has to be balanced against either the limited durability of bioprosthetic valves or the thromboembolic/bleeding risks from anticoagulation and mechanical valves. Earlier placement of bioprosthetic valves is likely to increase the proportion of patients who will require a second procedure due to the valve failing. Firstly, structural valve deterioration occurs earlier in younger patients [69]. Secondly, if earlier surgery does increase life expectancy, then a higher proportion will live long enough for failure to occur. Thirdly, the rate of reinvention will increase if earlier thresholds for intervention are also applied to the replacement valve.

Redo surgical AVR has traditionally been associated with significant mortality (10.3%) and morbidity [70]. Although redo surgical AVR remains a serious undertaking, the associated mortality has decreased. The latest national database report by the Australia and New Zealand Society of Cardiac and Thoracic Surgeons revealed a mortality of 4.0% for redo AVR compared with 1.2% for initial AVR [32]. Open heart surgery has been the mainstay method to deal with prosthetic valve failure, however other options are appearing. Series that have investigated TAVI valve-in-value (ViV) procedures for failing prosthetic valves have found comparable early mortality as initial surgical AVR, though with higher resultant valve gradients [71,72]. TAVI in TAVI is also an option that is starting to be explored [73]. When a valve replacement fails, the patients’ myocardium is again exposed to stresses that may cause further myocardial damage. An early AVR strategy may theoretically increase patients’ exposure to such myocardial injury.

Mechanical aortic valves carry the risks associated with life-long warfarin therapy, which many patients find unpalatable. A Californian state study from 1996 through 2013, showed that while warfarin successfully ameliorates the risk of stroke from mechanical aortic valves, there is a significantly higher risk of bleeding (approximately double control cohort rates) [74]. The burden of warfarin therapy can be somewhat lessened by patient self-managed warfarin using point of care INR machines. This method improves not only convenience but also decreases thromboembolic risk: a Cochrane review showed that it decreases thromboembolic risk (RR 0.47, 95% CI 0.31 to 0.70) compared to clinic based care but does not decrease the risk of major bleeding (RR 0.95, 95% CI, 0.80 to 1.12) [75].

While earlier AVR may improve long-term mortality, there are intervention risks regardless of the chosen method.

### 10.2. Infective Endocarditis

Infective endocarditis is an important complication following surgical AVR or TAVI, as shown by two recent nationwide registry studies in Finland and Denmark [76,77]. The Finnish study found an 8 year cumulative infective endocarditis risk of 1.28% for TAVI and 1.39% for surgical AVR [76], while the Danish study found a 5 year cumulative risk of 5.8% for TAVI and 5.1% for surgical AVR [77]. Both studies found no significant difference in infective endocarditis risk between TAVI and surgical AVR despite the concern that TAVI involves implanting more foreign material than a surgical approach. The Finnish study showed infective endocarditis was associated with a one-month mortality rate of 37.7% which increased to 52.5% mortality at one year. An Australian study found that in patients who have bicuspid aortic valves (BAV), infective endocarditis was 1.6-fold more likely after AVR than before it [78]. This suggests that an early surgery strategy carries an increased risk of infective endocarditis.

### 10.3. Effect of Age

Patients who have BAV tend to present at a younger age for AVR than trileaflet patients [79]. Earlier surgery for BAV patients would have the likely consequence of either increasing the proportion who would require a redo procedure for a bioprosthetic valve or facing a longer period of warfarin treatment for a mechanical valve. If the timing of surgery is incorrect, younger patients potentially have more years of life to lose than older patients. This was illustrated by a Swedish, nationwide, cohort study of life expectancy following AVR [5]. The estimated loss in life expectancy was higher for younger patients: 4.4 years (95% CI: 1.5 to 7.2 years) versus 0.4 years (95% CI: 0.3 to 0.5 years) in patients <50 and ≥80 years of age, respectively [5]. Potentially, prevention of irreversible myocardial damage by earlier surgery may be particularly beneficial for younger patients. TAVI does appear to be feasible and safe in selected older BAV patients although with increased rates of pacemaker insertion and paravalvular leak compared to non-BAV patients [80]. However, a high proportion of BAV patients have relative contraindications for TAVI due to aortic root disease, cusp configuration, calcification pattern, AR or poor vascular access [81]. TAVI ViV may be an option for BAV patients who initially have a bioprosthetic AVR that later fails.

### 10.4. Increased Reliance on Imaging for Decision Making

The 2014 guidelines’ recommendation for waiting until the onset of symptoms provides a clear clinical endpoint for AVR. Earlier timing of AVR while patients are asymptomatic, increases the importance of interpretation of echocardiographic and other radiological studies. Imaging-based interpretation of aortic disease, particularly AR is challenging [82]. Current imaging methods, particularly involving echocardiography, demonstrate significant intra-observer and inter-observer measurement variation [83]. To warrant surgery, ensuring a clear, reproducible pattern of valve disease over time and possibly incorporating information from non-echocardiographic methods would seem sensible. Several echocardiographic and non-echocardiographic parameters have been suggested to add extra prognostic information [84]. As previously discussed, CMR has interesting potential. Left ventricular global longitudinal strain (GLS) has gained much attention in the literature as a method that may improve timing of aortic valve surgery by revealing myocardial dysfunction before irreversible dysfunction has occurred [85]. A challenge with GLS is that discriminating cut-off values vary widely between studies with values between −19% and −12.5% for AR [86,87] and between −18.2% and −12.1% for AS [88,89]. There is significant overlap between GLS values between individuals who have good or poor outcomes [90], so GLS cut-off values currently appear to be limited to providing incremental prognostic utility [86].

### 10.5. Heterogenous Nature of Aortic Valve Disease

When commenting on the recent RCT by Kang et al. on early surgery for chronic AS [8], Lancellotti and Vannan highlighted that 22% of patients in the conservative group never received surgery during the median follow-up of 6.2 years [91]. This illustrates the heterogenous nature of aortic valve disease progression. Adopting an early surgery strategy may expose this subgroup to the morbidity and mortality of surgery several years earlier than what may be safely possible. The counterargument is that while this subgroup did not exhibit symptoms for years, irreversible myocardial damage may have been accumulating compromising long-term survival as evidenced by others.

Any decision for earlier surgery also has to be balanced against whether the patient is approaching a transition point which could alter their procedure type. For example, delaying surgery may tip the balance from a mechanical valve to a tissue valve, or surgical AVR to TAVI.

An earlier timing of AVR strategy is only possible when patients present early to clinicians. Unfortunately, many patients first present with severe symptoms associated with marked myocardial damage. The chance to gain the benefits from early surgery has already passed. This leads to an increased emphasis on primary care to screen for cardiac murmurs before symptoms arise.

## 11. Future Directions

Several RCTs are underway to investigate the utility of early AVR for severe AS. These studies include AVATAR (Aortic Valve Replacement versus Conservative Treatment in Asymptomatic Severe Aortic Stenosis; Clinical Trials.gov number, NCT02436655), EASY-AS (The Early Valve Replacement in Severe ASYmptomatic Aortic Stenosis Study; NCT04204915), and ESTIMATE (Early Surgery for Patients with Asymptomatic Aortic Stenosis; NCT02627391). Early surgery in patients who have raised brain natriuretic peptide (BNP) levels is being investigated by DANAVAR (Danish National Randomized Study on Early Aortic Valve Replacement in Patients with Asymptomatic Severe Aortic Stenosis; NCT03972644). Early TAVI is being studied by EARLY TAVI (Evaluation of Transcatheter Aortic Valve Replacement Compared to SurveilLance for Patients with AsYmptomatic Severe Aortic Stenosis; NCT03042104). Patients who have MRI evidence of fibrosis are being examined by EVOLVED (Early Valve Replacement Guided by Biomarkers of LV Decompensation in Asymptomatic Patients with Severe AS; NCT03094143). It will take a few years for these RCTs to provide a more definitive picture on the utility of earlier intervention for AS. In the meantime, the studies published since 2015 provide arguably a clearer picture than those that were available when the 2014 guidelines were developed.

Unfortunately, a search of ClinicalTrials.gov database revealed no in-progress studies to investigate early surgery for AR or MAVD. Hopefully the interest in early AVR for AS will generate interest for conducting trials for AR/MAVD. Historical data suggest that survival after AVR for AR is less than after AVR for AS, when adjusted for age differences [6,7]. Therefore, potential life expectancy increases from an early surgery strategy may be greater for AR than AS.

Further imaging studies (such as CMR and GLS) and biomarker studies (such as BNP) are required to determine whether identifying patients who have early myocardial damage can allow AVR to be timed at a stage that avoids unnecessarily early surgery yet protects long-term survival by avoiding severe myocardial damage. This strategy will be flawed if in reality, the optimal time for AVR occurs before measurable myocardial damage.

With the large increase in TAVI procedures [30], there needs to be verification of whether TAVI should have the same timing recommendations as surgical AVR, especially as the progression of TAVI into low risk and hence, younger patients, continues to expand. The optimal timing for surgical AVR appears to be related to avoidance of irreversible myocardial damage. Considering the similar long-term survival profiles between TAVI and surgical AVR [24,25,26,27], it is reasonable to expect that the optimal timing for TAVI will be the same. However, this assumption should be tested by RCTs. EARLY TAVI is one such trial in progress.

Further research is also required to determine whether age and/or bicuspid aortic valves affects the progression of myocardial fibrosis, which could have ramifications for AVR timing.

## 12. Conclusions

The most recent AHA/ACC guidelines that examine timing of AVR were published six years ago. They recommend that AVR should occur when chronic AR or AS cause symptoms or clear echocardiographic evidence of left ventricular decompensation (for example, LVEF <50%). Undoubtedly, aortic valve surgery has clear benefits over conservative treatment when a patient reaches this point. When viewing these guidelines, it is important to recognize their context. For chronic AR, there was a persistent pattern of recommending that patients wait until they have a less favorable long-term outcome. The context has arguably changed with the remarkable decrease in the perioperative mortality for AVR to less than 1% in centers of excellence. This shifts the focus from perioperative mortality to factors that affect longer-term survival. For AS, most of the guidelines’ referenced studies did not use long-term survival as their outcome. Instead, they used event-free survival where surgery itself was part of the outcome. This outcome does not discern whether postoperative long-term survival is affected by timing of surgery.

Over the last six years, several studies have added extra information. They have produced a consistent narrative for both AR and AS, that before patients reach the criteria recommended by the current guidelines, there is a window when they can achieve better long-term survival from earlier AVR. The RECOVERY trial [8] was the first published RCT looking at the timing of surgical AVR for AS. Although only a small study, its findings support a possible survival benefit from earlier surgery. Larger prospective RCTs are required to confirm or refute the benefit of earlier AVR. Fortunately, such trials are underway but will take time to be completed. Meanwhile, there is sufficient evidence to warrant revisiting the 2014 guidelines’ recommendations regarding timing of AVR for chronic AR and AS. A shift away from waiting until patients develop symptoms would have the important ramification of an increased reliance on the reliability of imaging methods to measure valve disease severity and early signs of myocardial damage. Additionally, patients would face either a higher chance of requiring redo procedures during their lifetime, or an increased duration of anticoagulation therapy.

## Figures and Tables

**Figure 1 jcm-09-02694-f001:**
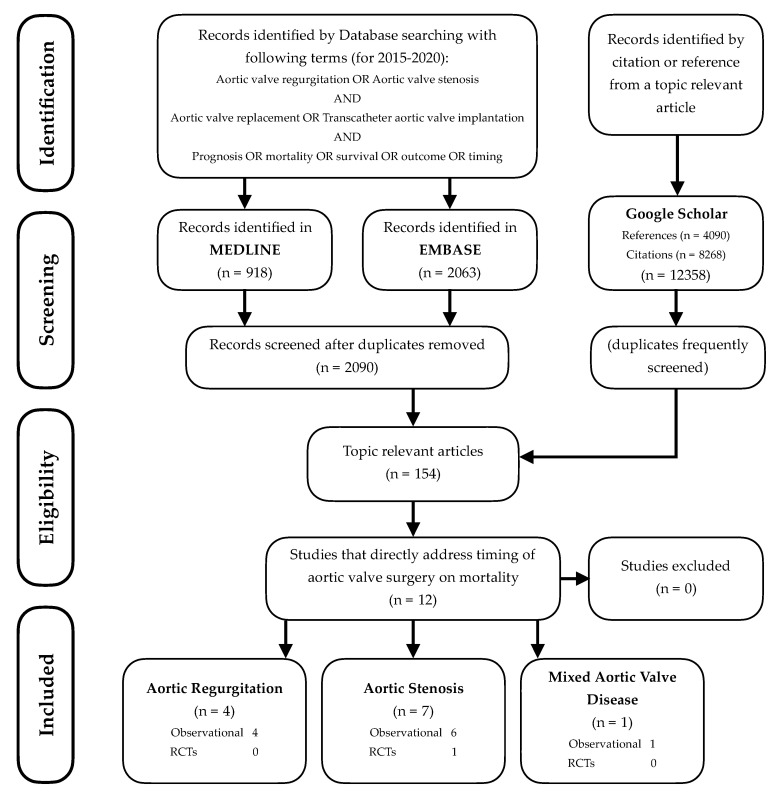
Flowchart of the study selection.

**Table 1 jcm-09-02694-t001:** Summary of AHA/ACC 2014 guideline recommendations for AR intervention.

Parameter	Details	COR	LOE
Symptomatic	AVR is indicated for symptomatic patients with severe AR regardless of LV systolic function (stage D)	I	B
LVEF < 50%	AVR is indicated for asymptomatic patients with chronic severe AR and LV systolic dysfunction (LVEF < 50%) (stage C2)	I	B
Other cardiac surgery, severe AR	AVR is indicated for patients with severe AR (stage C or D) while undergoing cardiac surgery for other indications	I	C
LVESD > 50 mm (LVESDi > 25 mm/m^2^)	AVR is reasonable for asymptomatic patients with severe AR with normal LV systolic function (LVEF ≥ 50%) but with severe LV dilation (LVESD > 50 mm, stage C2)	IIa	B
Other cardiac surgery, moderate AR	AVR is reasonable in patients with moderate AR (stage B) who are undergoing other cardiac surgery	IIa	C
LVEDD > 65 mm	AVR may be considered for asymptomatic patients with severe AR and normal LV systolic function (LVEF ≥ 50%, stage C1) but with progressive severe LV dilation (LVEDD > 65 mm) if surgical risk is low	IIb	C

Abbreviations: AHA/ACC, American Heart Association/American College of Cardiology; AR, aortic regurgitation; COR, Class of Recommendation; LOE, Level of Evidence; LV, left ventricular; LVEDD, left ventricular end-diastolic dimension; LVEF, left ventricular ejection fraction; LVESD, left ventricular end-systolic dimension; LVESDi, left ventricular end-systolic dimension index. Source: Nishimura et al. (2014) [11].

**Table 2 jcm-09-02694-t002:** Summary of AHA/ACC 2014 guideline recommendations for AS intervention.

Parameter	Details	COR	LOE
Symptomatic	AVR is recommended for symptomatic patients with severe high-gradient AS who have symptoms by history or on exercise testing (stage D1)	I	B
LVEF < 50%	AVR is recommended for asymptomatic patients with severe AS (stage C2) and LVEF < 50%	I	B
Other cardiac surgery, severe AS	AVR is indicated for patients with severe AS (stage C or D) when undergoing other cardiac surgery	I	B
Aortic velocity ≥ 5.0 m/s	AVR is reasonable for asymptomatic patients with very severe AS (stage C1, aortic velocity ≥ 5.0 m/s) and low surgical risk	IIa	B
Decreased exercise tolerance or exercise fall in BP	AVR is reasonable in asymptomatic patients (stage C1) with severe AS and decreased exercise tolerance or an exercise fall in BP	IIa	B
Dobutamine stress test, low-flow/low gradient severe AS	AVR is reasonable in symptomatic patients with low-flow/low-gradient severe AS with reduced LVEF (stage D2) with a low-dose dobutamine stress study that shows an aortic velocity ≥4.0 m/s (or mean pressure gradient ≥40 mm Hg) with a valve area ≤1.0 cm^2^ at any dobutamine dose	IIa	B
Low-flow/low gradient severe AS	AVR is reasonable in symptomatic patients who have low-flow/low-gradient severe AS (stage D3) who are normotensive and have an LVEF ≥50% if clinical, hemodynamic, and anatomic data support valve obstruction as the most likely cause of symptoms	IIa	C
Other cardiac surgery, moderate AS	AVR is reasonable for patients with moderate AS (stage B) (aortic velocity 3.0–3.9 m/s) who are undergoing other cardiac surgery	IIa	C
Rapid progression, severe AS	AVR may be considered for asymptomatic patients with severe AS (stage C1) and rapid disease progression and low surgical risk	IIb	C

Abbreviations: AHA/ACC, American Heart Association/American College of Cardiology; AS indicates aortic stenosis; AVR, aortic valve replacement by either surgical or transcatheter approach; BP, blood pressure; COR, Class of Recommendation; LOE, Level of Evidence; LVEF, left ventricular ejection fraction. Source: Nishimura et al. (2014) [11].

**Table 3 jcm-09-02694-t003:** Summary of studies that examine the timing of aortic valve surgery for chronic AR which were published subsequent to the release of the AHA/ACC 2014 guideline.

Author, Year, Location	Size	Type	Sample Details	Factor	Result		*p*-Value
Mentias et al. 2016 [14] Cleveland, Ohio, US	1417	Retrospective, observational	Age 54 ± 16 years; 75% male. Severe AR with LVEF ≥50%. 933 (66%) underwent AVR	AVR surgery during follow up	10 year survival:		<0.001
					Yes	87%	
					No	71%	
				Symptomatic	10 year mortality:		<0.001
					HR 2.06 (1.76–2.49) (compared to symptomatic)		
Murashita et al. 2017 [15] Rochester, Minnesota, US	530	Retrospective, observational	Age 57 ± 17 years; 80% male; 37% BAV All underwent AVR for severe AR	Symptomatic	10 year survival (CI):		<0.01
					Yes	77.8% (59.7–99.9%)	
					No	91.1% (85.7–96.6%)	
				LVEF	10 year survival (CI):		0.04
					≥60%	85.4% (81.7–89.2%)	
					<60%	69.5% (61.3–78.3%)	
				LVESD	Risk of left ventricular dysfunction at 1 year postoperatively, defined as LVEF below 60%:		<0.01
					>40 mm	odds ratio 5.39	
Yang et al. 2019 [16] Rochester, Minnesota, US	748	Retrospective, observational	Severe AR Age 58 ± 17 years; 82% male; 39% BAV 361 (48%) underwent AVR	Time-dependent AVR (within 6-months of initial echocardiogram)	Multivariate hazard ratio (CI) for all-cause mortality at median 4.9 years:		0.02
					0.36 (0.25–0.86)		
				Symptoms	Multivariate hazard ratio (CI) for all-cause mortality at median 4.9 years:		<0.0001
					3.16 (2.10–4.75)		
				LVESDi	Multivariate hazard ratio (CI) for all-cause mortality:		0.04 0.003
					<20 mm/m^2^	Reference	
					20–25 mm/m^2^	1.53 (1.01–2.31)	
					≥25 mm/m^2^	2.23 (1.32–3.77)	
de Meester et al. 2019 [17] Brussels, Belgium	356	Retrospective, observational	Age, 51 ± 15 years; 83% male; 42% BAV All underwent AVR for severe AR	Symptoms	10 year survival:		0.013 0.001
					NYHA class I	86 ± 4%	
					NYHA class II	73 ± 7%	
					NYHA class III/IV	65 ± 7%	
				LVEF	10 year survival:		0.011
					≥50%	80 ± 3%	
					<50%	69 ± 6%	
					Spline function analysis hazard ratio (CI) for cardiovascular events:		0.002
					≥55%	reference	
					<55%	4.13 (1.65 to 10.33)	
						per 1% decrease in LVEF	
				LVESDi	10 year survival:		<0.001
					<25 mm/m^2^	84 ± 3%	
					≥25 mm/m^2^	62 ± 6%	
				LVEDD	10 year survival:		
					No difference between LVEDD <65 mm and LVEDD ≥65 mm		
					No difference between LVEDD <70 mm and LVEDD ≥70 mm		

Abbreviations: AHA/ACC, American Heart Association/American College of Cardiology; AR, aortic regurgitation; AVR, aortic valve replacement; BAV: bicuspid aortic valve, HR, hazard ratio; LVEF: left ventricular ejection fraction; LVEDD, left ventricular end-diastolic dimension; LVESDi, left ventricular end-systolic dimension index; NYHA, New York Heart Association.

**Table 4 jcm-09-02694-t004:** Summary of studies that examine the timing of aortic valve surgery for chronic AS which were published subsequent to the release of the AHA/ACC 2014 guideline.

Author, Year, Location	Size	Type	Sample Details	Factor	Result		*p*-Value
Taniguchi et al. 2015 [18] Kyoto, Japan	1808	Multicenter, retrospective, observational	Asymptomatic, severe AS; age 77 ± 9 years, 40% male Propensity score-matched cohort of 582 patients	Initial AVR or watchful waiting	5-year survival:		0.009
					Initial AVR	84.6%	
					Watchful waiting	73.6%	
					5-year rate of hospitalization for heart failure:		<0.001
					Initial AVR	3.8%	
					Watchful waiting	19.9%	
Genereux et al. 2016 [19] New York, US	4 trials 2486 patients	Meta-analysis	Asymptomatic, severe AS	Early AVR or watchful waiting	All-cause mortality (CI):		0.01
					Early AVR	reference	
					Watchful waiting	3.7 (1.3–11.1)	
						fold higher	
Masri et al. 2016 [20] Cleveland, Ohio, US	533	Retrospective, observational	Asymptomatic, severe AS, LVEF ≥50%; age, 66 ± 13 years, 78% men, 31% with coronary artery disease	AVR or no AVR	Multivariable Cox proportional hazard survival analysis for 6.9 ± 3 years all-cause mortality (CI):		<0.001
					AVR	0.26 (0.16–0.41)	
					No AVR	reference	
				Exercise stress echocardiography % age-gender predicted METs	Long-term (6.9 ± 3.3 years) survival:		<0.001
					≥85%	85.0%	
					<85%	67.6%	
Lancellotti et al. 2018 [22] Liège, Belgium	543	Multicenter, retrospective, observational	Asymptomatic, severe AS; LVEF ≥ 50%; age 71 ± 13%; 61% male; all underwent AVR.	Peak aortic velocity	Survival at 2, 4, 6 years following AVR:		0.03
					<5.0 m/s	84 ± 2%, 78 ± 4%, 70 ± 6%	
					≥5.0 m/s	73 ± 8%, 65 ± 10%, 54 ± 13%	
				LVEF	Survival at 2, 4, 6 years following AVR:		0.02
					≥60%	87 ± 5%, 78 ± 4%, 69 ± 7%	
					<60%	67 ± 7%, 63 ± 8%, 63 ± 8%	
Campo et al. 2019 [21] Chicago, Illinois, US	265	Retrospective, observational	Asymptomatic severe AS.	Early AVR or watchful waiting	Survival at 2, 4 years:		0.033
					Early AVR	92.5%, 78.9%	
					Watching waiting	83.9%, 91.0%	
Kim et al. 2019 [23] Seoul, South Korea	468	Retrospective, observational	Asymptomatic, severe AS, LVEF ≥50%; age 64 years; 50% male. Early AVR was performed in 351 patients	AVR or medical treatment	All-cause mortality (median 60.9 months):		0.036
					AVR	9.1% per year	
					Medical treatment	2.4% per year	
					Hazard ratio 0.62 (0.40–0.97)		
Kang et al. 2019 [8] Seoul, South Korea (RECOVERY trial)	145	Prospective, single center, RCT	Asymptomatic, severe AS, LVEF ≥50%; age 64 ± 9 years; 36% male	Early surgery or watchful waiting (randomized)	5 year death from cardiovascular or surgery causes:		0.003
					Early surgery	1%	
					Watchful waiting	15%	
					5 year death from any cause:		
					Early surgery	7%	
					Watchful waiting	21%	

Abbreviations: AS, aortic regurgitation; AHA/ACC, American Heart Association/American College of Cardiology; AVR, aortic valve replacement; METs, metabolic equivalents; LVEF, left ventricular ejection fraction.

**Table 5 jcm-09-02694-t005:** Reported perioperative mortality rates from AVR over the last 30 years.

Year	Mortality	Author	Size	Location
1985	7%	Scott et al. [29]	1479	Stanford, California
2000	5.6%	Kvidal et al. [7]	2359	Uppsala, Sweden
2001	4.0%	Edwards et al. [28]	16,105	Jacksonville, Florida
2016	0.6%	Mentias et al. [14]	1417	Cleveland, Ohio
2018	0.9%	Lancellotti et al. [22]	1375	Liège, Belgium
2019	0.9%	Kim et al. [23]	468	Seoul, South Korea
2019	0.3%	Yang et al. [16]	748	Rochester, Minnesota

Abbreviations: AVR, aortic valve replacement/repair.

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
