# Peer review of "Early Valve Replacement for Severe Aortic Valve Disease: Effect on Mortality and Clinical Ramifications"

_jcm, 2020, doi:10.3390/jcm9092694_

Round 1
Reviewer 1 Report
Koerber et al. have reviewed the timing of aortic valve intervention for chronic AR and/or AS. A statistical meta-analysis was not performed, because of the published data is very heterogeneous. This paper provides valuable new insight to this clinically important issue.
Major comments:
Authors should discuss the role of prosthetic valve endocarditis as a potential disadvantage of the early AVR strategy. There are several recent papers demonstrating a relatively high incidence of endocarditis both after AVR and TAVI (Moriyama et al. Eurointervention 2019, Butt et al. JACC 2019).
Some of the tables are difficult to read and some of the data is missing. For example Table 4. Taniguchi et al. Column: Result Early AVR.., and Masri et al. Result No AVR.. In general, tables should be checked for clarity.
Author Response
We thank reviewer 1 for raising the issue of infective endocarditis following AVR/TAVI. It is an important consideration, and so a new section (10.2., lines 465-476) in the revised manuscript has been added to discuss its risk.
Table 4, Tanigushi et al. the result column for initial AVR was missing tabs, so the 84.6% was not in line with the 73.6% below it. This has been corrected.
Table 4, Masri et al. used a hazard ratio rather than providing absolute percentages. The “Early AVR” states “reference” to show that it was the baseline in the comparison.
Unfortunately, the source studies had a mixture of absolute rates (e.g. 84.6%) and relative rates (hazard ratios). This creates a busy table but is the nature of the data.
All tables have been rechecked.
Reviewer 2 Report
As a reviewer who is not specialized in this field, I believe the article provides a very relevant basis to re-evaluate current guidelines.
I believe it would be useful if the authors provided a summarized chart with the currently accepted recommendations (AHA/ACC) side-by-side with the findings from their evaluation.
In this chart, they could clearly state the changes they want to introduce.
They should also clearly state the extent to which their new observations will improve the current management.
It may be useful if the manuscript was shortened.
Author Response
We thank reviewer 2 for these comments.
Respectfully, we do not believe that it is appropriate for us to provide direct recommendations for clinicians as to how to manage their patients. Rather, this requires a dedicated independent committee who develop and update the guidelines. The main purpose of our manuscript is to provide an updated review of the literature that in turn argues that the guidelines require updating. We believe that in so doing, our manuscript will create discussion within the field. A summary table would certainly add clarity, but we do not wish to circumvent the role of governing organisations.
Regarding “state the extent to which their new observations will improve the current management”, it is tempting to look at tables 3 and 4 and state that an early AVR strategy seems to lead to a 10-20% reduction in 5 to 10-year mortality. However, we have avoided such statements because this would necessitate a formal meta-analysis. The data is non-randomised and is extremely heterogenous. It is interesting to note that all the identified studies found a benefit in the same direction - improved long-term mortality from earlier surgery and each individual study with its own design, found a statistically significant difference. We felt that it was more appropriate to comment on the qualitative direction of the results, which we have done throughout our paper.
Reviewer 3 Report
Congratulations. It is a well-prepared study and a
methodologically well-made article. It touches upon a key problem in the area of aortic valve surgery. It confirms the existing premises about a benefit of early surgical intervention in patients with chronic AR and AS and the impact on long-term
results. The article may have an important contribution to
tightening the criteria for surgical treatment of aortic valve diseases. Small remarks do not detract from the value of the publication.
-
In Prisma Flowchart: total number of studies included = 11; but 4 for AR, 7 for AS and 1 for MIX = it is 12???
-
The Chapter "Limited Durability of Bioprosthetic Valves or Risks from Anticoagulation" may raise some controversy: - We should remember that scenario (in some centers) - AVR Bio, next TAVI and VinV TAVI or TAVI next AVR Bio and VinV TAVI are not free of consequence including risk of re-intervention but their is the often overlooked fact that severe stenosis has been introduced several times in the same patient, which may not be indifferent to the heart muscle.
-
the issue of which option is the best for the patient and which prosthesis is the best, mechanical or biological, is still pending. we have to remember that perfectly prostheses, e.g. ONX and others, have a reduced INR in therapy, and we must remember about cost-effectiveness during the use of TAVI therapy in all patients (especially in low risk patients)
- In some regions - Europe TAVI is more clear than TAVR
Author Response
We thank reviewer 3 for their positive assessment of our manuscript and helpful feedback.
- Thank you for identifying the obvious miscount in the Prisma flowchart. The mixed aortic valve disease observational study was identified late during the manuscript preparation process and had not been properly included. The Prisma flowchart has been corrected to 12 and the abstract was also amended.
- A clearer statement (line 461) has now been added to convey the risks associated with re-do procedures. The idea that structural valve deterioration creates a possible second myocardial injury was also added in line 444.
- We agree that the decision between bioprosthetic, mechanical and TAVI is complex and still to clearly emerge from the literature. We have tried to avoid making any such recommendations in this review. We have tried to convey the idea that there are significant risks no matter which intervention is utilised, and an early AVR strategy may increase exposure to these risks.
- The Journal of Clinical Medicine is based in Switzerland so the terminology of TAVR has been replaced with TAVI. TAVI is also the more common terminology in the authors country, Australia.